# Faithful Model Evaluation for Model-Based Metrics

**Palash Goyal[*], Qian Hu[*], Rahul Gupta**
{palashg, huqia, gupra}@amazon.com
Amazon Alexa AI

## Abstract

Statistical significance testing is used in natural language processing (NLP) to determine whether the results of a study or experiment are likely to be due to chance or if they reflect a genuine relationship. A key step in significance testing is the estimation of confidence interval which is a function of sample variance. Sample variance calculation is straightforward when evaluating against ground truth. However, in many cases, a metric model is often used for evaluation. For example, to compare toxicity of two large language models, a toxicity classifier is used for evaluation. Existing works usually do not consider the variance change due to metric model errors, which can lead to wrong conclusions. In this work, we establish the mathematical foundation of significance testing for model-based metrics. With experiments on public benchmark datasets and a production system, we show that considering metric model errors to calculate sample variances for model-based metrics changes the conclusions in certain experiments.

## 1 Introduction

In the field of natural language processing (NLP), continuous progress hinges upon the development of novel techniques that outperform existing ones. However, accurately assessing the effectiveness of these new techniques requires a comprehensive evaluation framework. Model evaluation serves as the foundation for assessing the performance and impact of NLP advancements. Significance testing is a crucial tool in the evaluation process, enabling us to derive accurate conclusions. It allows us to determine whether the obtained evaluation results hold significance or are merely coincidental.

As the cost of human annotation for evaluating models using deterministic metrics is substantial, there is a growing trend towards utilizing model-based metrics for evaluation purposes. *Model-*

---

[*]These authors contributed equally to this work.

*based* metrics evaluate the outputs of an NLP model using another machine learning model such as using toxicity classifier to evaluate the toxicity of the generated texts by a text generation model, while *deterministic* metrics evaluate the outputs of a NLP model using annotated ground truth labels. In significance testing, computing the confidence interval plays a pivotal role in reaching precise conclusions. The computation of this interval relies on the sample variance, which differs depending on whether deterministic metrics or model-based metrics are used. In the case of deterministic metrics, the sample variance corresponds to the variance of the collected samples. However, for model-based metrics, where results are predicted by machine learning models, the sample variance is influenced by the model's prediction errors. Existing works using model-based metrics for model evaluation do not consider prediction errors for significance testing, risking inaccurate conclusions. In this work, we establish the mathematical foundation of significance testing for model-based metrics.

We conduct several experiments on using model-based metrics including hate speech detection (Hartvigsen et al., 2022) and user perceived defects detection. The experimental results show that considering prediction errors in significance testing changes the conclusions in certain experiments. Thus, we propose that the research community utilizes our framework for performing statistical testing with model-based metrics. In the following sections, we derive the mathematical equations for significance testing for model-based metrics and conduct experiments on several public benchmark datasets and a production system. Finally, we discuss related works and draw final conclusion.

## 2 Model Evaluation with Model-Based Metrics

In this section, we provide background on significance testing and then derive how we can modify

it to incorporate model's prediction errors when using model-based metrics.

## 2.1 Background - Significance Testing

Significance testing is a statistical analysis used to estimate the relationship between two statistical variables. When evaluating two models, we want to know if the performance of the two models is significantly different. In this work, we assume the model evaluation is a binary classification task such as whether the classified domain is correct or not in domain classification task, or the generated text is toxic or non-toxic in toxicity classification task. Given two models $C$ and $T$, their outputs are evaluated using a deterministic metric. The evaluation results are as following,

$$f_{C,1}, f_{C,2}, ..., f_{C,N_C}, \tag{1}$$

and

$$f_{T,1}, f_{T,2}, ..., f_{T,N_T}, \tag{2}$$

where $f_{C,.}, f_{T,.} \in \{0, 1\}$ are the evaluation results for outputs generated by models $C$ and $T$ respectively; $N_C$ and $N_T$ are the number of samples used to evaluate models $C$ and $T$. Their performance is estimated as the mean of the results,

$$\bar{f}_C = \frac{\sum_{i=0}^{N_C} f_{C,i}}{N_C}, \tag{3}$$

and

$$\bar{f}_T = \frac{\sum_{i=0}^{N_T} f_{T,i}}{N_T}. \tag{4}$$

The variances of the mean of evaluation results for $C$ and $T$ using deterministic metric are

$$\text{Var}^D(C) = \frac{1}{N_C(N_C - 1)} \sum_{i=1}^{|C|} (f_{C,i} - \bar{f}_C)^2, \tag{5}$$

$$\text{Var}^D(T) = \frac{1}{N_T(N_T - 1)} \sum_{i=1}^{|T|} (f_{T,i} - \bar{f}_T)^2, \tag{6}$$

respectively, where $D$ represents deterministic metric.

We want to know if their performance is significantly different, which is formally stated as a null hypothesis $H_0$ and an alternate hypothesis $H_a$:

$$
\begin{aligned}
H_0 &: \bar{f}_C = \bar{f}_T, \\
H_a &: \bar{f}_C \neq \bar{f}_T
\end{aligned} \tag{7}
$$

According to the central limit theorem, two sample means are statistically different if the following

symmetric confidence interval does not contain 0 (Smithson, 2003).

$$(\bar{f}_d - z_{\frac{\alpha}{2}} \sqrt{\text{Var}^D(d)}, \bar{f}_d + z_{\frac{\alpha}{2}} \sqrt{\text{Var}^D(d)}), \tag{8}$$

where $z_{\frac{\alpha}{2}}$ is the critical value and $\alpha$ is the confidence level, $\bar{f}_d = \bar{f}_T - \bar{f}_C$, $\text{Var}^D(d) = \text{Var}^D(T - C) = \text{Var}^D(C) + \text{Var}^D(T)$ (for the case $C$ and $T$ are dependent, the formula is derived in Appendix A.3). For 95% confidence level, $z_{\frac{\alpha}{2}} = 1.96$.

## 2.2 Significance Testing with Model-based Metrics

For model-based metrics, the performance of models $C$ and $T$ are evaluated by a metric model $M$, which is usually a statistical model with prediction errors. Thus, the sample variances calculated by Equation 5 and 6 are the variances of the observed evaluation values instead of the true evaluation values. In this section, we derive the sample variance considering the prediction errors.

Note that the following equations apply to both models $C$ and $T$. Assume we have $N$ independent and identically distributed (IID) samples of evaluations, let $N_+^O$ be the random variable denoting the number of observed positive samples. As we assume a binary classification task, each observation observes Bernoulli distribution. Therefore, the random variable $N_+^O$ observes binomial distribution with success probability $p^O$ (which can be estimated by using Equation 3 or 4). Thus, we have

$$N_+^O \sim Bin(N, p^O). \tag{9}$$

We aim to estimate the variance of distribution for the real positive samples, $N_+^R \sim Bin(N, p^R)$. Towards this goal, we derive the probability $p^R = P(R = 1)$ as following

$$
\begin{aligned}
p^R &= P(R = 1|O = 1)P(O = 1) \\
&\quad + P(R = 1|O = 0)P(O = 0) \\
&= p^{R|O} p^O + p^{R|O'} p^{O'}
\end{aligned} \tag{10}
$$

where $p^{R|O}$ and $p^{R|O'}$ are precision and false omission rate, respectively, which can be estimated from the metric model $M$'s performance on its testing data. The variance of a binomial distribution is $Np(1 - p)$, therefore, the variance of $N_+^R$, $\text{Var}^M(N_+^R)$ is

$$N(p^O p^{R|O} + p^{R|O'} p^{O'})(1 - p^O p^{R|O} - p^{R|O'} p^{O'}), \tag{11}$$

where $M$ represents model-based metric.

The variance of the model performance is

$$\text{Var}^M \left( \frac{N_+^R}{N} \right) = \frac{\text{Var}^M(N_+^R)}{N^2} = \frac{p^R * (1 - p^R)}{N}.$$
(12)

Since the population mean is unknown and variance is estimated with sampled mean $p^O$, the above estimator is a biased estimation. The corrected unbiased estimation using Bessel's correction (So, 2008) to account for the decreased degree of freedom is

$$\text{Var}^M \left( \frac{N_+^R}{N} \right) = \frac{p^R * (1 - p^R)}{N - 1}.$$
(13)

Therefore, the 95% confidence interval for model-based metrics is

$$(\bar{f}_d - 1.96\sqrt{\text{Var}^M(d)}, \bar{f}_d + 1.96\sqrt{\text{Var}^M(d)}),$$
(14)

where $\bar{f}_d = \bar{f}_T - \bar{f}_C$, $\text{Var}^M(d) = \text{Var}^M(T - C) = \text{Var}^M(C) + \text{Var}^M(T)$. Note that if metric model is perfect, the variance and confidence interval becomes the same as equations 5, 6 and 8. For proof, see Appendix A.1. Note that the formula can be easily extended to multi-class case (see Appendix A.2).

## 3 Experiments

We perform several experiments on public benchmark datasets and a production system to validate the proposed framework. In this section, we first introduce the experimental details on public benchmark datasets and then describe the experiments on the production system. Finally, we report the experimental results and analysis.

### 3.1 Experiments on Public Datasets

We select toxicity detection in natural language generation as the base task. The goal of this task is to detect if the generated text is toxic using a toxicity classifier. We adopt a state-of-the-art toxicity classifier, RoBERTa-ToxiGen (Hartvigsen et al., 2022) to detect toxicity in the generated text. We estimate precision and false omission rate (FOR) of this classifier on the manually annotated test set from ToxiGen. The estimated precision and FOR are 0.8897 and 0.22769, respectively.

We compare two text generation models GPT2 (Radford et al., 2019) and GPT-Neo (Black et al., 2021). To generate the text, we utilize prompts from BOLD (Dhamala et al., 2021) and RealToxicityPrompts (Gehman et al., 2020). BOLD (Dhamala

et al., 2021) is a manually curated dataset for bias measurement in open-ended language generation, which consists of 23,679 English text generation prompts for bias benchmarking in five domains including profession, gender, race, religion, and political ideology. RealToxicityPrompts (Gehman et al., 2020) has 100K naturally occurring, sentence-level prompts extracted from a large corpus of English web text.

#### 3.1.1 Result Analysis

Table 1 shows the experimental results. In the table, we show average toxicity score, average treatment effect (ATE), variance and confidence interval. ATE is calculated as the difference between average toxicity score of the two models, specifically, it is the average toxicity score of GPT-Neo subtracted by the average toxicity score of GPT2 (we consider GPT2 as the baseline).

From the table, we can see that there is a significant increase in variance when we consider the metric model errors. For example, on BOLD dataset, the variance of GPT2 changes from 1.92e-7 to 7.50e-6 (39x increase in variance). Disregarding the metric model errors, the confidence interval is (-0.00325, -0.00114), leading to the conclusion that we can reject the null hypothesis and reaching the conclusion that GPT-Neo produces output with significant lower toxicity than GPT2. However, when we consider the metric model errors, the confidence interval is (-0.00978, 0.00538), which shows insignificant difference and we cannot reject null hypothesis. In this case, considering metric model errors changes the final conclusion. On RealToxicityPrompts dataset, we also see big difference in variance change, but the conclusion is not changed.

### 3.2 Experiments in Production System

Besides conducting experiments on public models and benchmark datasets, we also perform experiments on live traffic in a production system of a lead voice agent. The task is natural language understanding such as domain classification, intent classification, etc. We compare the performance of two NLU models estimated by a machine learning model based on customer utterances and system responses (Gupta et al., 2021). The precision and FOR of the metric model are estimated on manually annotated datasets. The dataset used for experiment is de-identified.

Table 1: Experiment Results on Public Benchmark Datasets.

| Dataset | $\text{Mean}_{\text{GPT2}}$ | $\text{Mean}_{\text{GPTNeo}}$ | ATE | $\text{Var}^D_{\text{GPT2}}$ | $\text{Var}^D_{\text{GPTNeo}}$ | $\text{CI}^D$ | $\text{Var}^M_{\text{GPT2}}$ | $\text{Var}^M_{\text{GPTNeo}}$ | $\text{CI}^M$ |
|---|---|---|---|---|---|---|---|---|---|
| BOLD | 0.00456 | 0.00236 | -0.00219 | 1.92e-7 | 9.97e-8 | (-0.00325, -0.00114) | 7.50e-6 | 7.46e-6 | (-0.00978, 0.00538) |
| RealToxicityPrompts | 0.09124 | 0.09157 | 0.00033 | 8.34e-7 | 8.37e-7 | (-0.00220, 0.00286) | 2.06247e-6 | 2.063405e-6 | (-0.00365, 0.00431) |

[1] In the table, $\text{Mean}_{\text{GPT2}}$ means the average toxicity score of GPT2 model. Similarly, $\text{Mean}_{\text{GPTNeo}}$ is the average toxicity score of GPT-Neo model.
[2] $\text{Var}^D_{\text{GPT2}}$ means the variance of of GPT2 model using deterministic metric. $\text{Var}^M_{\text{GPT2}}$ means the variance of the GPT2 model using model-based metric.
[3] $\text{CI}^D$ means confidence interval using deterministic metric, and $\text{CI}^M$ means confidence interval using model-based metric.

Table 2: Experiment Results in a Production System.

| Experiment | $\text{Var}^D_C$ | $\text{Var}^D_T$ | $\text{CI}^D$ | $\text{Var}^M_C$ | $\text{Var}^M_T$ | $\text{CI}^M$ |
|---|---|---|---|---|---|---|
| Exp1 | 7.29e-5 | 1.01e-4 | (-0.05291, -0.00116) | 1.82e-4 | 2.04e-4 | (-0.06556, 0.01148) |
| Exp2 | 1.03e-5 | 1.00e-5 | (0.00007, 0.01777) | 1.07e-5 | 1.05e-5 | (-0.00010, 0.01794) |
| Exp3 | 4.93e-9 | 3.92e-9 | (0.00009, 0.00047) | 1.64e-8 | 1.63e-8 | (-0.00007, 0.00063) |
| Exp4 | 2.14e-8 | 2.11e-8 | (0.00006, 0.00086) | 6.77e-8 | 6.64e-8 | (-0.00026, 0.00118) |
| Exp5 | 8.84e-9 | 8.84e-9 | (0.000004, 0.00053) | 1.19e-8 | 1.19e-8 | (-0.00004, 0.00006) |

[*] The notation is similar as in Table 1.

### 3.2.1 Result Analysis

Table 2 shows the experimental results in a production system. The experiments are conducted on different NLU domains and devices. From the results, we can see that considering metric model errors has a big impact on variance estimation and also changes the final conclusion of the experiments.

## 4 Related Work

In their paper, Dror et al. (Dror et al., 2018) established fundamental concepts of significance testing in NLP research and proposed a simple practical protocol for statistical significance test selection in NLP setups. Berg-Kirkpatrick et al. (Berg-Kirkpatrick et al., 2012) investigate the relation between significance level and the magnitude of the gain. They also studied how the standard i.i.d. notion of significance hold up when there is domain shift. With the increasing of evaluation datasets, it is becoming common to evaluate NLP models on multiple datasets. Such multiple comparison poses challenges for significance testing in NLP. Dror et al. (Dror et al., 2017) proposes replicability analysis framework for the analysis of multiple comparisons between NLP algorithms. In their book, Dror et al. (Simpson, 2021) discuss the opportunities and challenges of using significance testing in NLP. The book covers multiple topics including choosing the appropriate significance test for a NLP task, dealing with the uniqueness of significance testing for non-convex deep neural networks, etc.

The current works on significance testing in NLP do not cover the area of model-based metrics. In this work, we propose to consider model prediction errors for more faithful evaluation and reaching the right conclusion when conducting significance testing for model-based metrics.

## 5 Conclusion

Significance testing is an important tool for us to draw accurate conclusions for evaluating NLP models. Existing evaluation works using model-based metrics do not consider model variance for significance testing, which can lead to wrong conclusions. In this work, we lay the mathematical foundation of significance testing for model-based metrics. We conduct experiments on public benchmarks and a production system. The significance testing results on these experiments show that model based errors need to be considered and incorporated for accurate evaluation.

In this work, we focus primarily on computing confidence interval with model-based metrics which use binary classification. In the future, we plan to extend our work to more general types of model-based metrics. Further, we assumed that the samples are independent and identically distributed. In practice, we often have a score associated with the metric model which can be used to relax this assumption. We leave this as future work.

### Limitations

This work mainly focuses on significance testing of binary categorical metrics and two sample t-test. We do not explore other types of metrics and statistical tests. We leave them to future work.

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

## A Appendix

### A.1 Sample Variance When the Metric Model is Perfect

Equation 13 is the variance when considering metric model prediction errors. When the metric model does not make errors, the precision and false omission rate is 1 and 0, respectively. Therefore, $p^{R|O} = 1$ and $p^{R|O'} = 0$. Substituting the values into Equation 10, we get $p^R = p^O$. Therefore,

$$\text{Var}^M \left( \frac{N_+^R}{N} \right) = \frac{p^O(1 - p^O)}{N - 1}. \qquad (15)$$

Next, we prove that Equation 15 equals Equation 5. For Equation 5, we denote the number of 1s as $N_1$, the total number of samples as $N_C$, and $p^O = \frac{\sum_{i=0}^{N_C} f_{C,i}}{N_C} = \frac{N_1}{N_C}$. Equation 5 can be rewritten as following

$$
\begin{aligned}
&\text{Var}^D(C) \\
&= \frac{1}{N_C(N_C - 1)} \sum_{i=1}^{N_C} (f_{C,i} - \bar{f}_C)^2 \\
&= \frac{\sum_{f_{C,i}=1}(1 - p^O)^2 + \sum_{f_{C,i}=0}(p^O)^2}{N_C(N_C - 1)} \\
&= \frac{N_1(1 - p^O)^2 + (N_C - N_1)(p^O)^2}{N_C(N_C - 1)} \\
&= \frac{N_1 - 2N_1 p^O + N_C(p^O)^2}{N_C(N_C - 1)} \\
&= \frac{p^O N_C - 2N_C(p^O)^2 + N_C(p^O)^2}{N_C(N_C - 1)} \\
&= \frac{p^O(1 - p^O)}{N_C - 1}.
\end{aligned} \qquad (16)
$$

Therefore, Equation 15 equals Equation 5 and 6. So Equation 13 is a generalized version of Equation 5 and 6, when considering metric model prediction errors.

### A.2 Application to Multi-Class Use Case

To apply the proposed approach to multi-class use case, we need to derive $p^R$ for multi-class classification tasks, which is as following

$$
p^R = P(R = 1) = \sum_{o_i \in \{0,1,...,N_O\}} P(R = 1|O = o_i)P(O = o_i) \qquad (17)
$$

where $o_i$ is the $i$-th class observation value. The variance of C and T can be derived using $p^R$.

## A.3 Variance When $C$ and $T$ are Dependent

In this section, we derive the variance of the difference for model-based metrics when $C$ and $T$ are dependent. When $C$ and $T$ are dependent, $\text{Var}^M(d) = \text{Var}^M(T - C) = \text{Var}^M(C) + \text{Var}^M(T) + 2\text{Cov}^M(C, T)$, where $\text{Cov}^M(C, T)$ is the covariance between $C$ and $T$. In the following, we derive $\text{Cov}^M(C, T)$.

$$
\begin{aligned}
\text{Cov}^M(C, T) &= \frac{1}{N}\text{Cov}^M(f_C, f_T) \\
&= \frac{1}{N}(E[f_C^R * f_T^R] - E[f_C^R][f_T^R]) \\
&= \frac{1}{N}(\sum_{x \in \{0,1\}, y \in \{0,1\}} xy \\
&\quad P(f_T^R = x, f_C^R = y) - p_C^R p_T^R) \\
&= \frac{1}{N}(P(f_T^R = 1, f_C^R = 1) - \\
&\quad p_C^R p_T^R),
\end{aligned}
$$

(18)

where $f_C^R$ and $f_T^R$ are the real values of a sample from $C$ and $T$, respectively.

The first term can be converted as following:

$$
P(f_T^R = 1, f_C^R = 1)
$$
$$
=
$$
$$
\sum_{x \in \{0,1\}, y \in \{0,1\}} P(f_C^R = 1, f_T^R = 1, f_C^O = x, f_T^O = y)
$$
$$
=
$$
$$
\sum_{x \in \{0,1\}, y \in \{0,1\}} P(f_C^R = 1, f_T^R = 1 | f_C^O = x, f_T^O = y)
$$
$$
P(f_C^O = x, f_T^O = y)
$$

Assuming conditional independence of $f_C^R$ and $f_T^R$ given $f_C^O = x, f_T^O = y$, we have the following,

$$
= \sum_{x \in \{0,1\}, y \in \{0,1\}} P(f_C^R = 1 | f_C^O = x, f_T^O = y) *
$$
$$
P(f_T^R = 1 | f_C^O = x, f_T^O = y) * P(f_C^O = x, f_T^O = y)
$$
$$
= \sum_{x \in \{0,1\}, y \in \{0,1\}}
$$
$$
P(f_C^R = 1 | f_C^O = x) P(f_T^R = 1 | f_T^O = y)
$$
$$
P(f_C^O = x, f_T^O = y)
$$

(19)

where $P(f_C^R = 1 | f_C^O = x)$, $P(f_T^R = 1 | f_T^O = y)$ are model performance metrics and can be estimated on a held-out dataset (similar to eq. 10 in Section 2.2) and $P(f_C^O = x, f_T^O = y)$ can be estimated empirically. Using Bessel's correction, we get

$$
\begin{aligned}
&\text{Cov}^M(C, T) \\
&= \frac{1}{N-1} \sum_{x \in \{0,1\}, y \in \{0,1\}} P(f_C^R = 1 | f_C^O = x) * \\
&P(f_T^R = 1 | f_T^O = y) * P(f_C^O = x, f_T^O = y) - \\
&\frac{1}{N-1}(p_C^R p_T^R)
\end{aligned}
$$

(20)