# OpenReview forum: "Faithful Model Evaluation for Model-Based Metrics"
_EMNLP/2023/Conference — EMNLP 2023 Main_

### Official Review · Reviewer_eZXe · 2023-08-02

**Soundness:** 4

**Excitement:**

4: Strong: This paper deepens the understanding of some phenomenon or lowers the barriers to an existing research direction.

**Paper Topic And Main Contributions:**

Statistical significance tests are used to determine whether the results of a study or experiment are likely to be due to chance or if they reflect a genuine relationship. However, in the recent years many experiments are evaluated using machine learning models. Those models introduce errors in the evaluations, and therefore, affect the confidence intervals that could derive to wrong conclusions. This paper addresses this problem and proposes a solution to take into consideration the errors produced by the evaluation model.

**Reasons To Accept:**

* This paper addresses a prevalent issue related to recent evaluations: evaluations based on another LMs may not be as accurate as using a gold standard.
* The proposed solutions seem a good drop-in replacement for the actual significant test.


**Reasons To Reject:**

* The proposed approach is parametrized by the evaluation model precision and false omission rate. However, those parameters are estimated with in-domain data, i.e., data from the same distribution as the data used for training the evaluation model. This may imply that the model performance on out-of-domain examples is not well estimated and therefore affect the variance estimation.

**Reproducibility:**

4: Could mostly reproduce the results, but there may be some variation because of sample variance or minor variations in their interpretation of the protocol or method.

**Reviewer Confidence:**

3: Pretty sure, but there's a chance I missed something. Although I have a good feel for this area in general, I did not carefully check the paper's details, e.g., the math, experimental design, or novelty.

---

> ### Author Rebuttal · Authors · 2023-08-28
>
> We want to thank the reviewer for the review and providing the suggestions that can improve the paper quality. Given the feedback, there is one concern. We address it below.
>
> 1. The proposed approach is parametrized by the evaluation model precision and false omission rate. However, those parameters are estimated with in-domain data, i.e., data from the same distribution as the data used for training the evaluation model. This may imply that the model performance on out-of-domain examples is not well estimated and therefore affect the variance estimation.
>
>     > We mentioned in the paper that the evaluation model precision and false omission rate are estimated on the evaluation dataset, not on the training data. Out-of-domain estimation is not considered in the paper. We can consider it as future work.

---

### Official Review · Reviewer_sWNL · 2023-08-04

**Typos Grammar Style And Presentation Improvements:** None
**Soundness:** 3

**Excitement:**

3: Ambivalent: It has merits (e.g., it reports state-of-the-art results, the idea is nice), but there are key weaknesses (e.g., it describes incremental work), and it can significantly benefit from another round of revision. However, I won't object to accepting it if my co-reviewers champion it.

**Missing References:**

None

**Paper Topic And Main Contributions:**

The paper provided a novel estimation method with regard to the variance of the evaluation results in model-based metrics and proposed a novel significance testing method based on the proposed variance estimator coupled with the two sample z test for comparing two NLP models according to model-based metrics.

**Questions For The Authors:**

(1) Is the proposed method suitable for precision, recall and F-score?


(2) The paper assumed that the evaluation results of $C$ and $D$ are i.i.d. Is the assumption too strict?

**Reasons To Accept:**

The idea is interesting. The mathematical expressions are straightforward and effective. The experimental results demonstrated the reasonability of the variance estimation proposed in the paper.

**Reasons To Reject:**

I consider it is unreasonable to decompose the variance $Var^M(d)$ into the form of $Var^M(C)+Var^M(T)$ because the evaluation results outputted by models $C$ and $T$ are frequently correlated. Moreover, I don't think the proposed method is suitable for the commonly used metrics, such as precision, recall, F-score.

**Reproducibility:**

5: Could easily reproduce the results.

**Reviewer Confidence:**

4: Quite sure. I tried to check the important points carefully. It's unlikely, though conceivable, that I missed something that should affect my ratings.

---

> ### Author Rebuttal · Authors · 2023-08-28
>
> We want to thank the reviewer for the review and providing the suggestions that can improve the paper quality. Given the feedback, there is a technical concern on an equation in the paper. We address it below.
>
> 1. I consider it is unreasonable to decompose the variance Var^M(d) into the form of  Var^M(C) + VarM(T) because the evaluation results outputted by models  and  are frequently correlated.
>
>     > In the paper, we make the assumption that C and T are independent. This assumption applies to several use cases such as online A/B testing. However, to address reviewer’s concern, we also relax the assumption and derive the covariance formula [here](https://www.dropbox.com/scl/fi/9d200jjqkff7y1sf8dvg4/FaithfulEvaluation_Supplementary.pdf?dl=0&rlkey=a8atyf12rt17n5zntkhfrhlnu). We will include this as an appendix in the camera-ready version. Note that we make a weaker conditional independence assumption that $f_C^R$ is independent of $f_T^R$ given $f_C^O$ and $f_T^O$. This assumption is weaker since we are not assuming that $C$ and $T$ are independent but if we know the observed values of both, the real values can be independently derived.
>
> 2. Moreover, I don't think the proposed method is suitable for the commonly used metrics, such as precision, recall, F-score.
>
>     > Similar to accuracy, to calculate the variance for precision and recall, we can use the derived $p^R$. Therefore F-1 score, as we can derive F-1 score from precision and recall.

---

### Official Review · Reviewer_ZfFZ · 2023-08-05

**Soundness:** 3

**Excitement:**

3: Ambivalent: It has merits (e.g., it reports state-of-the-art results, the idea is nice), but there are key weaknesses (e.g., it describes incremental work), and it can significantly benefit from another round of revision. However, I won't object to accepting it if my co-reviewers champion it.

**Paper Topic And Main Contributions:**

This paper introduces a significance test for model-based metrics. It lays out the mathematical foundation of this test and applies it to both public and production datasets. In both cases, the conclusions might be changed when the variance from model prediction is taken into consideration.

**Questions For The Authors:**

I don't have further questions besides asking the authors to respond to the previous section.

**Reasons To Accept:**

1. Very important topic given the current trend of using model-based metrics in more and more places.
2. Very clearly written and easy to understand. The experiments are solid and to the point.

**Reasons To Reject:**

1. Only tested two datasets and a few models. It would also benefit greatly from having model-based metrics with different levels of variance.
2. Only for binary classification. I concede that the authors acknowledged this point in the limitations section, but it's a bit odd given that section 2.1 is set up for a more general scenario.

**Reproducibility:**

4: Could mostly reproduce the results, but there may be some variation because of sample variance or minor variations in their interpretation of the protocol or method.

**Reviewer Confidence:**

3: Pretty sure, but there's a chance I missed something. Although I have a good feel for this area in general, I did not carefully check the paper's details, e.g., the math, experimental design, or novelty.

**Typos Grammar Style And Presentation Improvements:**

In Table 2, the superscripts seem to be inconsistent with previous notations: should've been D and M?

---

> ### Author Rebuttal · Authors · 2023-08-28
>
> We want to thank the reviewer for the review and providing the suggestions that can improve the paper quality. Given the feedback, there are two concerns and a minor question. We address them below.
>
> 1. Only tested two datasets and a few models. It would also benefit greatly from having model-based metrics with different levels of variance.
>
>     **Only tested two datasets and a few models**
>     > Our goal in the paper is twofold: 1) raise awareness in the community that using metric based models can lead to inaccurate conclusions, 2) establishing a mathematical foundation of significance testing when using metric based models. Our experiments on the two data sets covering two different tasks (toxicity detection and domain/intent classification) and two different systems (public benchmark and a production system) show that this issue is present and that using the proposed modifications in the paper we can correct these issues. Although performing experiments on more datasets would show further applicability, respectfully, we believe experiments on the selected datasets demonstrate the key point of the paper.
>
>     **Different levels of variance**
>     > The experimental results show that the variance of the model-based metrics spans a wide range. For example, in the production system, the variances range from as low as 1.19e-8 to as high as 2.04e-4.
>
> 2. Only for binary classification. I concede that the authors acknowledged this point in the limitations section, but it's a bit odd given that section 2.1 is set up for a more general scenario.
>
>     > The proposed approach is generally applicable to multi-class classification if the classes are independent of each other. In the paper, for simplicity and brevity, we show the derivation for a binary classification use case. We derived the formula for multi-class use case as the following and we’ll add it to the camera-ready.
>
>     > ---
>     >To apply the proposed approach to multi-class use case, we need to derive $p^R$ for multi-class classification tasks, which is as following
>     > \begin{equation}
>     > \begin{split}
>     > p^R = P(R=1) = \sum_{o_i \in \{ 0, 1, ..., N_O \} } P(R=1|O=o_i) P(O=o_i)
>     > \end{split}
>     > \end{equation}
>     > where $o_i$ is the $i$-th class observation value. The variance of C and T can be derived using $p^R$.
>
> 3. In Table 2, the superscripts seem to be inconsistent with previous notations: should've been D and M?
>
>     > Thank you for pointing this typo out. We’ll fix this for the camera-ready version.

---

### Meta-Review · Area_Chair_6pgg · 2023-09-10

**Recommendation:** 5

**Metareview:**

This paper introduces a significance test for model-based metrics. This topic is crucial in NLP given that many experiments are evaluated using machine learning models which can errors in the evaluations, and  affect the confidence intervals that could lead to wrong conclusions.
The paper is clear and well written. While the experiments are conducted only on few tasks and binary classification tasks, a generalization of the formula is proposed that could applied to multi-class classification tasks.

---

### Decision · Program_Chairs · 2023-10-07

**Decision:**

Accept-Main

**Comment:**

This paper introduces a significance test for model-based metrics. This topic is crucial in NLP given that many experiments are evaluated using machine learning models which can errors in the evaluations, and  affect the confidence intervals that could lead to wrong conclusions.
The paper is clear and well written. While the experiments are conducted only on few tasks and binary classification tasks, a generalization of the formula is proposed that could applied to multi-class classification tasks.